# Qualitative Focus Groups with Professionals of Special Education and Parents of Young Females with Intellectual Disability Exploring Experiences with Menstrual Hygiene Management and the Trigger for the Non-Therapeutic Hysterectomy in Mexico

**DOI:** 10.3390/healthcare10091690

**Published:** 2022-09-04

**Authors:** Maria del Rosario Flores-Medina, Edith Valdez-Martinez, Horacio Márquez-González

**Affiliations:** 1Medical Research Unit in Clinical Epidemiology, National Health Research Council, Mexican Institute of Social Security, Mexico City 06720, Mexico; 2Department of Clinical Research, Children Hospital ‘Federico Gomez’, Mexico City 06720, Mexico; 3Congenital Heart Disease Department, Cardiology Hospital, Centro Médico Nacional SXXI, Mexican Institute of Social Security, Mexico City 06720, Mexico

**Keywords:** qualitative research, intellectual disability, menstrual hygiene, hysterectomy, adolescent, child, Mexico

## Abstract

How primary carers, physicians, health education professionals, and others see or understand the subject of menstruation in women with intellectual disability (ID) is rooted in the socio-cultural context and in the socio-economic structures in which all of them live. The aim of this study was to explore how parents of young females with ID and special education professionals perceive and experience menstrual hygiene management, which coping strategies are applied; and what triggers the performance of a hysterectomy. A qualitative focus group study design was conducted with 69 parents and 11 special education professionals, in 14 schools and one Down syndrome clinic, in Mexico City. Data were analysed using the method of thematic analysis. The main concern of parents was how to cope with the underlying disease. They perceived menstrual bleeding positively. Their psychological distress had to do with the reproductive health of their daughters, with their wish to avoid pregnancy, and with their fear of death and leaving their daughters alone and helpless without them. None of them favoured hysterectomy. Medical indication of hysterectomy was identified as the trigger for its performance. There is an urgent need of policy development/review on best practices for hysterectomy in the females in question.

## 1. Introduction

For the purposes of this study, girls and young females with intellectual disability (ID) refer to people with significantly reduced ability to understand new or complex information, to learn new skills (impaired intelligence), and to cope independently (impaired social functioning), which started before adulthood and has a lasting effect on development [1].

Although the 11th version of the International Classification of Diseases and Related Health Problems of the World Health Organization (WHO-ICD-11) [2] replaces the WHO-ICD-10 conceptualization of intellectual disability with the concept of disorders of intellectual development. The health services and health statistics officers of most countries in the Americas still apply WHO-ICD-10, which uses the classification of “mild, moderate, severe or profound mental retardation” to denote ID. This classification takes into account the criterion of the intelligence quotient to define impaired intelligence, the evidence of impaired social functioning, and limitations in the individual’s daily activities and self-care skills [2].

A 2011 meta-analysis [3] of studies published between 1980 and 2009 shows that the prevalence of ID, across all 52 studies included, is higher in studies based on low- and middle-income group countries (16.4 and 15.9 per 1000 population, respectively) and in studies based on children/adolescents (0.4 and 1.0 per 1000 population). Although the prevalence of ID in children/adolescents is not high, these figures can be alarming—particularly in girls and young women with ID—given the limitations in available resources, in such countries, to manage ID and the lifetime impairment of cognitive and motor functions that make menstrual hygiene management difficult.

In that regard, a 2018 systematic review of the literature [4] reveals that hysterectomy to cope with menstrual hygiene is still a live issue in high-, middle-, and low-income countries. In high-income countries, it is performed with authorization from the court; while in low- and middle-income countries (such as Mexico), there is no active involvement of the state or financial or training support for women with ID and their carers. In the included studies, physicians perceived hysterectomy as a safe procedure and a solution for women with ID, whose menstrual hygiene is problematic. A 2019 systematic review of the literature [5] shows a lack of menstruation training, information, and support provided to people with intellectual impairments and their carers; high costs of menstrual products and a lack of appropriate options for people with physical impairments.

In Mexico, a recent cross-sectional, retrolective chart review of anatomical pathology and hospital records shows that—in the young females with ID—the mean age of having a hysterectomy is 15 ± 2.9 years, that it is performed predominantly in women with ID moderate or severe, that prophylactic-total abdominal hysterectomy was the most frequently performed; and, that the surgery is performed without the involvement of the clinical ethics committee of the respective hospitals.

The most frequent medical reasons to indicate hysterectomy are fertility control, management of menstrual hygiene, and risk of sexual abuse [6]. There are no studies that report lay perception regarding menstrual hygiene management in females with ID, neither at the ‘individual level’ (e.g., perceptions of menstrual hygiene) nor at the ‘social level’ (e.g., social creations of menstrual hygiene management). The majority of the lay perceptions, at the individual level, having been reported are from studies conducted in developed countries [4] or some other developing countries (e.g., India, Taiwan) [4].

Data from the Mexican national diagnosis of the situation of people with impairment [7] and the Mexican national measurement of poverty [8] show that the full exercise of human and social rights is not assured for women with ID; the main causes are low access to and poor quality of education, limited accessibility and availability of healthcare, and the fact that psychology and psychiatry services have never been a high priority at hospitals; to these causes, the high rate of people with an impairment that live with an income below the poverty line must be added. The main effects of these causes are reported to be low human development and the consequent fact that women with ID depend on their parents or family members and high episodes of discrimination and violence against them. The problematization of reality implies that ID is not a simple condition but a complex result of social reasoning.

The purpose of this study is therefore to explore these issues in depth. The study is both timely and relevant. The evidence generated is urgently needed to understand the reality of the situation of females with ID and to provide the basis for designing efficient interventions that combat the health inequities and enhance the well-being of girls and young women with ID in low- to middle-income countries such as Mexico.

The objective of the current study was to explore how parents of young females with ID and professionals of special education perceive and experience menstrual hygiene management, which coping strategies are applied, and what triggers hysterectomy performance.

## 2. Materials and Methods

### 2.1. Study Design

From June 2019 to January 2020, a qualitative focus group study design was conducted in schools of special education in Mexico City. This qualitative research method is particularly well suited for exploring sensitive matters, allowing for a deeper insight into feelings, thoughts, preferences, experiences, and practices, of the participants [9].

### 2.2. Recruitment and Research Participants

The study consisted of a total of 15 focus groups with a mean of 10 ± 5 participants in each group. The groups’ discussions were held with 11 professionals of special education and 69 parents of 69 female young women with ID. The participants were recruited from 14 schools (ten public and four private) attended by girls and young females with ID. These schools were selected (as described below) from the National Directory of Associations of and for People with Disabilities (2002), published by the National Institute of Statistics and Geography (INEGI, its acronym in Spanish) [10]. From the list of all 88 schools registered in the mentioned directory, four schools (two public and two private) were initially randomly selected, the same that were accepted to participate. Subsequently, based on the evolution of the analysis of the information generated, it was necessary to develop eight more focus groups that included primary carers from eight other schools that were selected by snowball sampling. At the end of the study, three more focus groups were organized and held with primary carers from other two schools and from a Down syndrome clinic. The goal of selecting those schools and a Down syndrome clinic was to hear from people with a wide range of experiences in menstrual hygiene management.

*Focus group procedures.* One of the authors (RF) communicated to the responsible person or the principal of the schools, via telephone, to invite each of them to participate in the research project. Afterwards, the letter of approval of the research ethics committee together with a written invitation to participate was sent to the responsible person or the principal of the schools that were interested in participating; it included a brief explanation of the study. Later on, each of them held several face-to-face interviews with one of the authors (RF) to ensure that he/she understood the means and methodologies required to set up the focus groups and undertake the research process. As a result, they were the ones who identified all the primary carers that met the inclusion criteria and then they invited them to participate in the focus groups.

### 2.3. Terminology

The group of **parents** comprised the persons directly responsible for the care of the young women with ID: 61 mothers, four fathers, three grandmothers, and one aunt. The group of **professionals** (professionals of special education) was made up of six teachers, two social workers, one psychologist, and two women whose job was to support the young women’s menstrual hygiene at the schools.

### 2.4. Sample and Sampling

Purposive sampling with maximum variation [11] was used to select parents and professionals. It was oriented to include different characteristics of the participating individuals, for example, differences in age, gender, education status, job category, geographical zone of residence, and type and modality of the school, thus ensuring a rich source of data. “The sample size was determined using the Sandelowski criterion [12]. The criterion states that in a qualitative study, the sample size is not determined a priori but is based on the robustness of the theory that is generated. A robust theory is one that is sufficiently deep and wide to be able to explain the knowledge generated by the research. Following this criterion, saturation was reached after 12 focus groups, the last three did not yield any new information”. For the characteristics of the participants see Table 1 and Table 2.

### 2.5. Pre-Focus Group Activities

The research project was evaluated and approved by the National Research Ethics Committee of the Mexican Institute of Social Security, and given registration number F-CNIC-2018-133. Access was granted by heads of schools of special education, both private and public, in Mexico City and in the State of Mexico. Written informed consent was obtained from all participants prior to the focus group discussion in which they participated. The anonymity of the information was guaranteed by tagging the data in a manner that could not lead back to the participating individual. 

One of the authors (RF) was the moderator of each of the focus groups on the basis of an interview topic guide. Another member of the research team (GC) was also present during each focus group; she was in charge of writing the field notes. Both of them were trained in the focus group technique. When it was corroborated (by means of a pilot test) that they had mastered the focus group technique for generated data, the field study began.

### 2.6. Procedure and Data Collection

An interview topic guide (“Appendix A”) was developed based on theoretical knowledge and group discussions with the research team. The guide underwent adaptations throughout the study, as a function of the analysis of the information being generated and new threads of questioning being identified. It also included probe questions to better understand participants’ responses to the discussion questions.

The topics explored were broad and intended to help the participants describe and discuss their experiences in as little or as much detail as they wanted. The guide incorporated topics related to values, beliefs, preferences, interests, ideas, experiences, and practices associated with menstrual hygiene management for females with ID.

Demographic data were collected before the focus group began.

The meetings for the focus groups were held in a private enclosure, at the site, date, and time selected by the participants.

The focus groups were conducted in Spanish and lasted a median of 60 min (range, 32–117 min). All discussions were audio recorded. A research assistant (GC) transcribed all the recordings *ad verbatim*. The researchers independently reviewed each of the recordings with the respective transcriptions in order to corroborate the correct emphasis of each of the arguments.

### 2.7. Analysis

The analysis took place simultaneously in the data generation process, allowing refinement of the topic guide and later data collection (as an iterative process). The data/narratives were analysed by using thematic analysis [13,14]. This means that, first, the data/narratives were read and re-read closely, in an active way, initially to corroborate the transcriptions with the audio and later to make notes of relevant aspects and meanings. Second, and being familiar with the data, meaningful fragments were coded (open coding) in a systematic way; through the coding process the codes were reviewed, and modified―if necessary. Third, themes or patterns within data/narratives set were initially identified from the analysis of the codes. Fourth, at this point, the themes developed were reviewed, and modified―as necessary. Finally, on a fifth phase, the data within each theme, and the connection among themes, were again analysed and refined―if necessary.

These five phases of the analysis involved constant, iterative, and reflexive revisions of each of the narratives, independently completed by at least two researchers (RF and EV), independent of one another; any discrepancy was resolved by consensus. The researchers and the research assistant (GC) held various joint sessions throughout the study. The information generated by means of the focus groups was captured for its analysis and managed by using Atlas ti software (version 8.0. Atlas.ti Scientific Software Development Mnbh, Berlin, Germany). Inductive and deductive focuses were used to organize and analyse the information contained in each of the narratives.

## 3. Results

The synthesis of the 15 focus groups revealed four interconnected themes: (1) problems or circumstances that make it difficult to deal with menstruation; (2) feelings, beliefs, and experiences; (3) strategies to deal with menstrual hygiene management; and (4) hysterectomy (their values, preferences, and beliefs).

The examination of each theme for its contribution toward understanding how participants perceive and experience menstrual hygiene management, which coping strategies are applied, and the triggers of hysterectomy performance, along with the analysis of the cohesion and inter-relations between themes according to this rationale, resulted in a thematic map (Figure 1).

The themes are described below with anonymised quotations; in brackets appear the number corresponding to the focus group.

### 3.1. Problems Identified with Menstruation and Its Management

#### 3.1.1. The Underlying or Base Disease as a Problem

The participating parents of nine focus groups [1,2,3,4,7,8,9,11,13] agreed that the main “*problem*” they faced was the underlying disease of their daughters. In four focus groups, there were parents of girls and adolescents with severe and moderate ID who said they had experienced *“great fatigue”* and *“stress”* while raising their daughters [1,4,8,9]. In addition, some parents identified problems resulting from the underlying disease: *“Lack of free time”* [2,4,9]; *“Difficulty finding someone reliable to take care of [their daughters] to have free time”* [2,8]; *“Little support from relatives”* and *“Even having the support, we continue with the obligation of care”* [1,2,4,7,8,9]; also, the *“Difficulty of not having a partner with whom to live and cohabit”* [7].

Some highlighted the importance of *“Acceptance of the underlying disease”* as the key point to improving the quality of life, both for them as parents and their daughters with ID [1,2,4,7,8,9]. They described acceptance of the disease as *“A very difficult process”* [1,2,7,9], *“Long”* [1,2,9], *“Heavy”* [1,2,4] and *“Tired”* [1,4,8,9]. For some of them, this process entailed *“Frustration”* [1,9], and *“Struggle”* [1,9]. One mother illustrated how she had achieved acceptance of her daughter’s underlying disease:


*“Currently, I feel very happy, very satisfied with my daughter… obviously, in the beginning, it is like a struggle that one has to go through, until acceptance. When my daughter was seven years old, she entered the children’s psychiatric hospital, I could not talk to my daughter… I cried, cried and cried, so I received psychological therapy until I was able to accept my daughter, and when I succeeded, I said: well, that’s it!… And what did I do? Well, take care of her… Starting from there, I had to work for it” [P1, MG].*


Some of the participating professionals [P2, P3] mentioned having observed that acceptance of the underlying disease represents *“**A challenge”*, and *“A titanic job”* to the parents.

#### 3.1.2. The Adolescent’s Ability to Understand

Parents of daughters with moderate and severe ID agreed that *“It has been very difficult to deal with”* the deficiencies caused by the underlying disease [1,2,4,7,8,9,10,11,13,14], mainly due to: “*The lacking or poor understanding*” that their daughters achieve from the verbal information provided [2,4,9,10,11,13,14]; the “*Difficulty they have to express their discomforts and needs*” [1,2,11,13,14]; and “*For their inability to perform certain self-care tasks*” [1,2,4,10,11,13,14]. Circumstances/factors that they (as parents) identified as barriers to menstruation management. For instance:


*“It is hard to understand her… she tells us one thing, she specifies something, and for us, it is something like that on the air… very frustrating” [P1, PJ].*



*“She has not a properly developed language; then, making her understand that menstruation is coming and that it is completely natural is a problem; because there is no way to explain it to her directly, as you would do with someone who has a more extensive language” [P13, MK].*


#### 3.1.3. The Use of Sanitary Pads 

The *“Rejection”* of the sanitary pad [6,10] and/or the *“Difficulty”* to negotiate their use [11] were associated with the absence of pre-menarche education on the subject of menstruation, in adolescents with mild and moderate ID. For instance:


*“She did not like it, she made this [moves hands in the form of rejection] to take it away, she did not want it” [P6, M2].*



*“She comes home and takes off her pad… and says, with an annoyed expression: ‘This is getting on my nerves. And I insist: ‘Baby, you have to wear that’, and she says ‘No’” [P11, MS].*


Regarding adolescents with mild and moderate ID who did accept the use of pads, some parents mentioned that, for them, the problem with the use of pads was that their daughters *“Want to change the pad very often”* [14,15]; and for others, the problem was that their daughters *“Want to be with the same pad all day”* [10,11].

#### 3.1.4. Behaviour Disturbances 

An “*Important problem*” highlighted by some parents of women with mild and moderate ID was that they (their daughters) are not aware of the intimate nature with which menstrual bleeding should be managed: *“They announce or discuss their menstrual bleeding with other family members and/or in public places”* [7,11] or “*They take off their pads in public*” [9,11] and/or *“They don’t close the bathroom door while using it”* [7]. This conduct was indicated as “*Worrying*” and “*Stressful*” and “*Shame conditioner*”. To illustrate this, one mother explained: “*I am looking for a way for her to take care of herself, to close the door, to be careful… we have been talking a lot; since she has a cognitive level of a three or four-year-old girl*” [P7, 25M].

In this respect, the statements of the group of professionals point out that: “What we have had to see at school… is that suddenly the girl comes out showing off her pad all over*…” [P6, 24]*.

Another behaviour problem discussed, particularly by parents of adolescents with moderate ID, was that of “*They show indiscriminate and impulsive affection with strangers*” *[P14, P15].*


*“My daughter is tremendous… She is always hugging people, she hugs and kisses everyone, and she doesn’t care, even men… and that scares me a lot. I always talk to her and tell her: ‘No, daughter…’ It is very difficult for me to make her understand, that is my concern more than anything” [D15, MA].*


Another behavioural difficulty, negative emotions expressed during the period of menstrual bleeding, and even days before it, came up in the discussion of 10 focus groups [1,2,4,5,8,9,10,11,12,13]. The negative emotions expressed were such as: *“Rage”*, *“Anger”*, *“Fear”*, *“Crying”*, *“Sadness”*, and *“Disgust”*. As well, the fact that “*They do not identify what it is*”, “*They do not realise what is happening*”*,* or “*They cannot express I have pain or it is coming*”. A situation that in particular the parents of adolescents with severe and moderate ID defined as *“Very emotionally complicated days”*, *“Very difficult days”*.

#### 3.1.5. School

Many parents of adolescent girls who attended public schools said they preferred not to take them to school during the period of menstrual bleeding [2,6,8,11,12,14]. One reason cited by some of them was the *“Pain”* [6,12,14]; however, the majority agreed that it is *“Very complicated”* having to go to school to perform menstrual hygiene management for their daughters [2,8,12]. The parents who agreed to take their daughters daily to school were those whose daughters are partially or totally independent in menstrual hygiene management [10,12].

Within the context of participating in public schools, the professionals validated the absenteeism of adolescents during the period of menstrual bleeding. One teacher said:


*“They do not bring them regularly, they leave them at home, precisely because the girls have not been helped with managing their cycle… that is why most choose not to send them [to school]. Parents only let us know: ‘They are in their menstrual cycle, and teacher, the truth is, I prefer not to bring her because I am the one who has to be cleaning her’. Parents who decide to take their daughters to school must sign [a document, at school] that they will be on the phone for when the girls have to go to the bathroom because they have to go change them themselves…” [P3, L].*


In private schools [4,7,9], parents reported preferring to take their daughters to school during the period of menstrual bleeding. Except for one, who stated, “*She is not going to leave the house those days, because I prefer to have her under my care, so that she does not expose herself*” *[P9, M6].* The teachers affirmed that “*There is no absenteeism problem related to menstrual bleeding… Parents are not required to help or assist their daughters with pad changes and hygiene management*” [4].

### 3.2. Feelings, Beliefs, Experiences

In all groups, menstrual bleeding was perceived positively. This was defined as: *“A natural process”* [4,6,8,9,10,11,13,15], *“Something normal”* [2,4,5,7,10,11,14], *“Something biological”* [6], *“A sign that the body is working well”* [6]. There were mothers of adolescents with severe ID and dependent on menstrual hygiene management who stated:


*“Menstruation is not something that important in our daughters, because, for the problems that one has, menstruation is the least of it… The fact that she menstruates, that she stains are the least of it” [P1, MG, MK, MJ].*


Regarding menarche, in ten focus groups [2,4,6,8,9,10,11,13,14,15] the parents of adolescents who were already menstruating (regardless of the degree of ID) agreed on their description of the experience during the first episode of menstrual bleeding. This generated a series of emotions in them, which they themselves labelled as feelings of: *“Fear”*, *“Concern”*, *“Terror”*, *“Stress”*, *“Physical shock”*, *“Nostalgia”*. For example:


*“I did get scared, and the truth is, I then felt in my body that I couldn’t touch her. I didn’t have sisters, and well, we’re not ready, right? One only cries” [10, PZ].*



*“For me, it was a shock (…); yes, a responsibility, a great responsibility, for the care that we must have, not only in hygiene… Once the girl begins her menstrual period, she is subject to pregnancy…“ [10, ME].*


In this respect, the emotions and feelings of parents of girls without menarche were pointed out. To illustrate, two mothers described: 


*“I’m not going to lie to you, I’m very scared, I don’t know what will happen, I think I start crying with her [she has an anguished expression] because literally, I’m alone [denotes that she wants to cry, but she contains herself]. I recently separated from my husband; so, the truth is, I am very scared; I have nightmares, I dream that my daughter is raped, that she becomes pregnant, that I see her give birth [she says it very quickly, she has an expression of anguish and terror]; I mean, literally (…), the truth is, I’m very scared” [11, M9].*



*“I feel that she is going to take it normally, but, well…, I’m afraid that moment will come, I imagine it will be very difficult…” [2, M1].*


Stance on pregnancy. Most of the informants (except three mothers) from ten focus groups [2,4,6,7,8,9,11,12,14,15] agreed on the importance of avoiding pregnancy in the women in question. This assumption prevailed among parents and special education professionals, regardless of ID grade.

For parents, their daughters: “Would not have the ability to take care of another being” or “They would not have enough maturity or the necessary tools to bring the little one forward”. In addition, some exclaimed: “You end up taking care of the babies of disabled women because they are neither capable nor responsible”. Many of them supported ideas such as: “The high chances of bringing a baby with a disability”. Some of them added: “I don’t know how long I will live”, and “One will not be eternal”. Others assured that: “She is not going to have sex, while we live with her, we are going to take care of her”.

There were three mothers, within the same discussion group, who defended their idea of respecting the autonomy of their daughter with ID: *“She is going to decide what happens to her body. She can do her things by herself… I know that maybe we have to help or support her if she decides to have a child… I couldn’t take that away from her just because I didn’t want to help her…”* [2,8,14].

Although the professionals agreed with the parents on the importance of avoiding pregnancy in the women in question, they highlighted the woman’s right to: “*Live her sexuality*” and “*Live her life as a couple*”.

The thoughts of the parents’ future absence and who might be responsible for their daughters after their death was an emerging theme in the group discussions [2,6,7,8,11,15]. All participants talked a lot about the “*Fear*”, “*Panic*”, “*Anguish*”, and “*Sadness*”, which generates the uncertainty of “*Who is their daughter going to stay with*” and “*What could happen to their daughters*”. As one parent explained:

*“But I am distressed, I have woken up with anxiety and panic attacks; yes, I have woken up like that… Because I think, what will happen when I die [he has an anguished expression]. Even though I think my son is going to treat her well, I am in a hurry to know whom she is going to stay with. This, too, was the most pressing issue for my wife when she died; furthermore, my wife made my sister swear that if no one would take care of her [she refers to her daughter with ID], she would take care of my daughter… [he has an expression of pain and sadness when he remembers this fact]; So yes, I do feel anguish, and a lot [you can see the tension in his face]…”* [7].

Only in one group [7] the parents, who were elderly, had already considered taking actions, such as: “*The search for legal guardians and the contracting of life insurance*”. A mother was echoed, in her discussion group, regarding her proposal to “*Make them as independent as possible, to prepare them for the future*” *[8, M2].*

### 3.3. Strategies to Face Menstrual Hygiene Management

#### 3.3.1. Support Network

Parents frequently discussed the value and importance for them of being supported (in a tangible way in the care and management of menstrual hygiene) by their relatives: eldest daughter(s), sister(s), mother or mother-in-law, and/or husband [4,6,9,10,11,12,14,15]. As one mother highlighted: “*My husband helps me, he knows how to put on the pad, he tells her: ‘Put it on like this…’. The moment the girl wants to go to the bathroom, he goes with her and he just checks…, picks up her pad, gives her the clean pad… and he checks her*” *[6, M2].*

All mothers highlighted the importance of emotional support for them from male relatives and the school. This emotional support was described in terms of empathic relationships and demonstrations of affection and love towards their daughter with ID. One mother described: “*M**y husband helps me, he arrives in the afternoon, he takes care of her, from the time she arrives until the time the girl sleeps… He gives her her medications, takes care of her*…” *[14, MT].*

“Frustration”, “Overwhelm”, and “Disappointment”, were the feelings expressed in the parents’ discussions regarding the total or partial lack of a support network [1,5,6,7,9,11,15]. In this regard, a mother illustrated how difficult it is to resolve or alleviate the need for family support, despite the use of monetary strategies: *“I have to pay my granddaughter or my daughter or some other relative; because… I don’t feel the support of my family, not really… Sometimes I say to my granddaughter or my sister, for example, ‘Stay with the girl… I have to go to the doctor on Sunday, stay with her, right?’ And they answer ‘No granny, I don’t have time’ or ‘No sister, I can’t’. And I only answer them, ‘Don’t worry, it is ok.’ Who do I tell!” [7, M].*

Schools, mainly private ones, were recognized as a “*Place of support*” by special education professionals and parents [1,2,4,9]. The professionals considered the school as substantial support for parents, as it is the school that supports them with information through workshops and suggestions for menstrual hygiene management; in addition to guiding adolescent girls with ID during pad changes: “*We coordinate with the parents, we offer them the support they require; for example, keeping track of them when going to the toilet and that they are taking care of their hygiene*” [2].

Several parents acknowledged the support of special education professionals and expressed appreciation: “*… The smell is not pleasant… As the teacher says ‘It beats you down…’ So, I appreciate their help [of the teachers] when my daughter has her period*” *[1, MG].*

#### 3.3.2. Menstrual Hygiene Management Training 

All the parents, particularly mothers, spoke a lot, during the group discussions, about their experiences in the educational process they used to train their daughters in menstrual hygiene management. Although many mothers declared to have touched on this subject before menarche, the majority did so until the first menstrual bleeding appeared.

The proportion of girls informed or trained before menarche on the subject of menstruation and/or menstrual hygiene management was higher in those with mild ID [2,8,9,10,11,13,14,15], followed by those who had moderate ID [11,13,15]. Most of the girls and adolescents with severe ID wore diapers and were not *de facto* educated or trained in menstrual hygiene management; the parents asserted that their daughters “*Would not understand*”.

All mothers claimed to be the only resource for training and guidance. All of them denied having received any type of specialized training regarding the management of intellectual disability and/or menstrual hygiene management [2,8,9,10,11,13,14,15]. The educational process they used was based on their own experience.

Most mothers opted for the strategy of bathing together with their daughters: “My girl is 10 years old; she has not started yet… What I do right now is take her to the bathroom with me, I try to explain how it will be for her when she menstruates…” [15, MA]. Other mothers, particularly those with dependent daughters or daughters with moderate ID, reported that, in addition to constant verbal explanation, they used dolls or illustrations: “I use a doll, and I say, for example: ‘We are going to put her pad on her or we are going to change it or we are going to take it off…’” [9, A]. Others made their daughters wear protective panty liners: “My girl is 11 years old, she still does not menstruate, but I have already explained it to her… She has even started to wear protective panty liners because she has seen me use sanitary pads… And now, I already tried to explain it. In addition, to my understanding, to my ignorance, I have told her that it was a woman’s secret, that she should not tell anyone… And yes, she accepted it” [15, MP].

Many mothers said they had waited until menarche to start ongoing training on the hygienic and private management of menstrual flows.


*“She knows what menstruation is… When changing the pad, I show it to her and say: ‘Look… does it go on like this?’ Likewise, when I take off the pad, I teach her what she should do… Also, in some way, I tell her: ‘Don’t say that [she refers to menstrual bleeding], you have to go to the bathroom and call me, but don’t tell people [she raises the volume of her voice]. No one should know that you have your period.’ But she has a cognitive level of a girl of three to four years, so, imagine that…” [7, M].*



*“Right now, it’s only three months [she refers to the menarche], I still have to be saying: ‘Look, this is a pad, we have to put it here…’. I have to be careful that she doesn’t get dirty, to take her to the bathroom… I know that she is going to learn it in time, and she is going to do it; but, well, it’s going to take a little while…” [14, MT].*


In public schools [2,3,5,6,8,10,11,12,13,14], unlike in private schools, *teachers* limit themselves to verbally reminding the adolescents to change their pad and insist to the parents about the need for them to train/teach their daughters on menstrual hygiene management. *“**Parents are told:* ’*Teach her [to do it] every four hours’; sometimes we do that too… Parents are also told: ‘Please do it the same way as you do for toilet training’” [13, MT4].* Nannies only verbally guide adolescent girls: *“When I accompany a girl to the bathroom, I assist her, I just say: ‘Take it off’; that is, without touching her, from afar: ‘Take it off, put it in the bag’, and that’s it… Parents are always advised that our role is to guide them in changing their pads, but superficially, we can’t go too far either” [3, L].*

Psychologists and social workers give educational talks to parents and implement workshops related to human sexuality, on the topics of gender and respect for minorities, prevention of sexual abuse and sexual violence: *“We talk about sex education in the orientations and in the workshops that we offer directly to parents. We always talk about self-care, setting limits, that not everyone can touch her body, that not everyone can be supporting her in using the bathroom” [6, P1].*

### 3.4. Hysterectomy

#### 3.4.1. Beliefs

None of the professionals spoke of hysterectomy as a means of the menstrual hygiene management. However, all of them agreed that girls and adolescents with ID should protect themselves against unwanted pregnancies. For instance: *“**The surgery [refers to salpingectomy] is just so that they do not get pregnant, but not as such to eliminate their period” [5, M4].*

All professionals agreed that non-therapeutic hysterectomy carries more risks than benefits. One of them explained: *“… It is a total alteration because her body is not prepared for a situation like this… If the uterus is removed, menopause will come earlier…” [2, L1]. One more added: “The advice I always give to moms is: if you operate her, do not tell anyone [taps the table]. Come on, not even the father! Because it’s like putting a ‘There are no pending matters here, abuse her… it could happen’. You don’t tell anyone that you operated her…’” [5, D].*

Although all parents showed concern for the reproductive health of their daughters, none of them recognised hysterectomy as a means of menstrual hygiene management [1,2,3,4,6,8,9,11,12,14,15]. All of them considered hysterectomy: (a) as an extreme solution [1,4,6,9,11,15]: “*We felt like we were going to mutilate our daughter, and we said no, it’s not fair, we better take care of her, the time will come when she reaches menopause and then we remove the uterus*” *[1, MG].* “*I think it is an extreme solution, come on, we would not do it to ourselves, even less to them” [9, MA].* (b) As violating their rights [8,9]. When asked what rights they were referring to, they mentioned the “*Pain*”. To illustrate this, here is an excerpt of what was mentioned by a mother: “*… Violate her rights… What rights? Her rights to decide, because I am exposing her to that pain…*” *[9, M6].* (c) As “*Something that could affect her life*” [15, MA]. (d) As blocking a natural process: “*Hysterectomy, that is, you remove an organ that is not going to be functional, and the problem is over [she says surprised]; yes, but you are also blocking a natural process” [9, M7].* Additionally (e), as infringing her willingness, this last assertion was made in particular by mothers of daughters with Down syndrome [11,14,15].

#### 3.4.2. Information Processing and Choice

All parents asserted that the proposal to perform the hysterectomy was presented to them (by the doctors treating their daughters) as the only alternative [1,2,3,4,6,8,9,11,12,14,15]. The reasons given by the treating physicians were in terms that intellectual disability places adolescent girls in a state of vulnerability for sexual abuse and the risk of unwanted pregnancies.


*“The pediatrician immediately told me: ‘You know what, why are you thinking about it? I mean, the only possibility you have with her is to remove the uterus. Medically, what is recommended for families is to operate them due to the situation that they are very vulnerable girls, they are very innocent girls…’ ‘The pediatrician also told me: ‘I understand that you take care of your daughter and you are always on the lookout for her; but… well, you never know…’*



*I thought they [surgeons] had to open it like a cesarean section or something like that, but she told me ‘no’, it’s through the navel… She even told me: ‘Look, you know that the uterus is only for having children, it does not help you to do anything other than having children, and the only thing you would do is help your daughter… in case of sexual abuse. And I have to tell you (she told me), I advise you to learn more about the operation, they are very vulnerable girls… Obviously, do it before it arrives’ [she refers to the menarche…]” [4, MY].*


And in terms of what is stipulated in the Penal Code of Mexico City. To illustrate this, the mother of a 15-year-old girl with moderate ID stated: *“The gynaecologist told me that the current legislation in Mexico City prohibits performing sterilization methods on this group of women; but, because she is a patient with mental retardation and has abundant menstruations, the best thing would be to perform a total prophylactic hysterectomy”* [15]. This is what was said by another mother who had a 12-year-old girl, without menarche, and with moderate ID: “*The gynaecologist read to me what is stipulated in Article 151 of the Penal Code and explained to me that it was impossible to carry out any sterilization procedure because the Penal Code prohibits it; but that due to my daughter’s mentally impaired conditions, she would not have the capacity to take adequate hygiene measures, so she recommended a total hysterectomy*” [15].

Many of the parents did not know what was hysterectomy or the “Surgery” for their daughters: *“I had understood it as a tubal ligation; that is, cutting the tubes “ [2, AF]. “I would like more than anything to put her through surgery for planning…” [15, MP].*

The professionals [2,5,12] agreed with what was indicated by the parents regarding the pressure exerted by doctors to perform the hysterectomy: “*Doctors tell parents: ‘There is no other option, if I leave her like this, without operating her, she will have a family and her problem will multiply’” [2, TS]. “Doctors often begin to put pressure on parents, saying ‘You have no other alternative…’” [2, L1].*

## 4. Discussion

### 4.1. The Underlying or Base Disease

Parental experiences revolved around coping with the underlying disease. Menstrual bleeding was not perceived as a bigger issue than coping with the underlying disease; even the embarrassment caused by the lack of ability of young females with ID to take care of themselves during the menstrual bleeding was an undervalued issue by parents. The problematization of the parents’ reality implies that ID entities and impaired social functioning (e.g., emotional and behavioural difficulties) are not simply real but based on the type and course of the base disease, the parental idiosyncrasies, and the context or habitat in which they lived. To illustrate, data from a qualitative study [15] exploring the challenges that parents experience when caring for children with mental disorders reveals psychological and emotional challenges (stress by caring tasks and worries about the present and future life of their children), and feelings of sadness and inner pain due to the disturbing behaviour of the children. As well as social (burden of caring task, lack of public awareness and social support) and economic (child care interfering with various income generating activities in the family, extra-expenses associated with the child’s illness) challenges.

The fact that parents did not receive education in caring for the underlying disease of their daughters redounds in behaviour that mirrors incomprehension, which creates barriers to effective health and well-being for their daughters with ID. 

### 4.2. Menstruation and Education

The findings also indicated what a 2017 literature review [16] and various systematic reviews and meta-analyses [4,5,17] have shown in low- and middle-income countries that many young females start their menstrual periods uninformed and unprepared and that mothers or female relatives were the primary source of information and advice for young females regarding menstruation, menstrual health, and hygiene. In this study, the training process (when it was conducted) included how to collect blood and disposal of materials and intimate and private handling. Parents reported not having received training from any source. This assertion is consistent with those reported in other studies in developing countries [4,5,16,17].

In examining the roles of the participating professionals as providers of menstrual hygiene education (for parents or young females), they ranked them as the less common sources of information. Their educational workshops for parents were focused on issues of gender, preventing sexual abuse, and respecting minorities. This provides evidence about the role that schools and professionals can play in providing information and awareness for parents of women with ID. This omission in effective menstrual hygiene education to parents is intensified in the arena of sexual and reproductive health. Previous literature identifies the lack of education (family, carers, and people with ID) on menstrual hygiene management [4,5] and sexual health education as the major problems [18,19] that are detrimental to the health and well-being of the main agents involved. 

### 4.3. Pregnancy

In this respect, the onset of menarche roused a mix of negative reactions from parents. Even in mothers of premenarcheal girls, when considering menarche in the near future, their fears and anxieties were intensified. Negative reactions were derived from the idea that menarche represents the begging of sexuality and an increasing vulnerability of a young female to sexual abuse and unexpected pregnancies rather than the burden of care of menstrual bleeding. Worldwide, sexual abuse is known to be especially common in individuals with ID [18,19], in particular children and young people with ID [18,20]. Moreover, sexual abuse may be difficult to detect in these individuals, as they may lack literacy and communication, or it may be assumed that they are not sexually active.

Many parents in this study and in other published study [21] have assumed that their daughters with ID will never become mothers. There were common assumptions and concerns among them that their daughters with ID cannot be responsible for the tasks of giving their children a good upbringing as well as the minimal support available, and that a mother’s disability may be inherited by the baby. So, some of them intentionally approached physicians seeking information or a reversible or permanent method of contraception.

Although parents discussed their anxieties and concerns regarding the possibility of their death prior to their daughters’ death and the uncertainty as to what will happen with the menstrual care and the fact of their vulnerability, all participants (parents and professionals) rejected hysterectomy. The professionals added further that hysterectomy increases the opportunities of exposing the young female to sexual abuse. In a 2021 systematic review of the literature [19], women with ID who had experienced sexual abuse stated that they were targeted because they had been sterilized and thus could not become pregnant because of the abuse.

Both parties posited that hysterectomy is an extreme solution and it blocks a natural process and concurred that it is not a solution. Other studies carried out in low- and middle-income countries share this view: a quantitative survey [22] and two qualitative studies [23,24].

Notwithstanding that, both parents and professionals declared that physicians persisted with the idea of non-therapeutic hysterectomy as a best practice for females with ID. They highlighted a popular saying of physicians: *“Uterus is unnecessary in women with ID if they will not become pregnant; on the contrary, it causes problems”*. This, on the one hand, conveys a symbolic violence toward women with ID; and, on the other hand, it pictures how physicians’ decisions are made in light of their own expectations, beliefs, values, and standards of quality of life. Across the literature, there are often discussions of physicians’ lack of experience in the population with ID [25,26,27,28].

Though treating physicians were voiceless in this research, the participating parents and professionals gave a clear explanation of the medical decision-making process. Two interconnected strands of justice are required to address this.

First, performing a non-therapeutic hysterectomy in girls or young females with ID has the intrinsic effect of harming (the patient); even if hysterectomy is presently considered a safe operative procedure, it violates the negative duty not to harm. More clearly, the physician-parent/patient relationship is fiduciary [29]. Parents believe and trust that each of the treating physicians for their daughters (based on the principle of non-maleficence) would do nothing to wrong and to harm them.

Second, the way in which physicians framed information disclosed to parents was misleading; moreover, they did not present key alternatives or objectives nor were they interested in knowing the parents’ preferences; hence, the parents were impeded to make a truly autonomous choice, and physicians were contributing to healthcare inequalities [21].

The professional medical dominance of decision-making along with poor parents’ health literacy and the consequent passive role of them in medical decision-making united the way in which the hysterectomy was framed by physicians (i.e., in terms of a better quality of life, a safe operative procedure, and, in some cases, a procedure legally accepted) are factors that mirror stigmatized societal views about intellectual disability. Multidisciplinary efforts to reduce the historically conceived inequity in women with ID must be strategically focused on building in-country capacity. The capacity-building issue should be understood as a continuous dynamic process of strengthening of abilities to perform care functions, solve problems, define and achieve objectives and understand and deal with development needs [30]. This goes beyond health system development. The capacity building must include an emphasis on the overall system―within its political context, the cultural environment or context within which individuals and organizations, together with the organizational governance mechanism and structures, interact to align their activities with their stated values and goals.

### 4.4. Strengths and Limitations

The study here reported has a number of strengths. This is the first study of its kind in Latin America. It provides a picture of feelings, values, and beliefs that revolve around menstruation and hysterectomy and that are shared by the direct carers of Mexican women with ID. It should be mentioned that extreme care was taken in the methodological rigour with which this research was performed in order to reduce potential biases that are characteristics of focus group discussions. The methods used and the active focus on the process of research that was carried out guaranteed the representativeness of the sample. In brief, the active focus of the research refers to the identification of the question that forced the researchers to think, to confirm or to refute, to gather more data, and to pursue emerging paths of research [13].

Nevertheless, there are some limitations. This study relies solely on focus group data from parents and professionals of special education. This could be seen as a limitation to the full understanding of the *emic* perspective on the Mexican culture—as we did not include more ethnographic techniques for data generation or multiple sources of data. Nonetheless, the fact that (a) the participating parents had different genders, ages, educational backgrounds, and kinship; (b) the participating professionals of healthcare education had different genders, ages, work experiences, and work categories; and (c) the women with ID had different ages, ID degrees, educational backgrounds, and were from urban or mixed rural-urban settings; together with the fact that we included private and public schools, provide a good foundation for developing a better understanding of the perceptions and experiences gained by participants on menstrual hygiene management and hysterectomy. It is also important to note that this study is not generalizable in the same sense as quantitative research because it involves a non-random, purposive sample of individuals who contributed to the generation of data.

Despite the limitations mentioned; and, by way of conclusion, this study is particularly useful and has clinical and social value because it highlights the urgency of incorporating ethical thinking into quotidian clinical practice in order to transform a physician−parent/female with ID relationship that is merely technical-bureaucratic into a relationship that is truly professional and personal; that is, at the service of the patients with ID. As well, the fact that the results showed that non-therapeutic hysterectomy is still a live issue in Mexico is indicative of the need to set up effective educational and training programmes on menstrual hygiene management, sex and sexual health, in the context of IDs entities; both at the individual (professionals of especial education, families, carers, and women with ID) and community level. This study also generates new material for a normative inquiry that attempts to develop (based on evidence) and to lead to the enactment of policies and statutes in this area of public health and clinical practice to fix this serious and long-standing problem.

## Figures and Tables

**Figure 1 healthcare-10-01690-f001:**
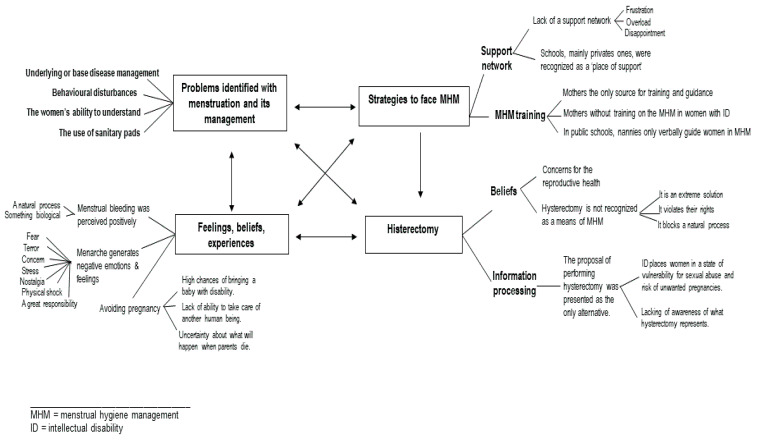
Thematic map of factors contributing to problematize the reality that is lived by women with ID, in the views of their parents and professionals of special education.

**Table 1 healthcare-10-01690-t001:** Demographic information of the 80 carers of the 69 young females with ID.

	^a^ Parents *n* = 69 Number (%)	^b^ Professionals of Special Education *n* = 11 Number (%)
**Carer’s age, mean (SD)**	44 (±10)	38 (±7)
Carer’ gender		
Female	65 (94)	9
Male	4 (6)	2
**Carer’s marital status**		
Married	37 (54)	
Single	19 (28)	
Widow/widower	7 (10)	
Living together	3 (4)	
Without data	3 (4)	11 (100)
**Carer’s relationship to women with ID**		
Mother	61 (88)	
Parent	4 (6)	
Grandmother	3 (4)	
Aunt	1 (1)	
**Carer’ residence**		
Mexico City	62 (90)	7 (64)
State of Mexico	7 (10)	4 (36)
**Carer’s highest education level ^c^**		
Basic schooling	22 (32)	
Medium superior level	25 (36)	1 (9)
Superior level	22 (32)	10 (99)
**Carer’s employment status**		
Housewife	38 (55)	
Employed (informal)	15 (22)	
Employment (formal)	13 (19)	^d^ 11 (100)
Retired	3 (4)	

^a^ Parents: 61 mothers, 4 fathers, 3 grandmothers, and 1 aunt. ^b^ Professionals of special education: 6 teachers, 2 social workers; 1 psychologist; and 2 women whose jobs were to support young women with menstrual hygiene, at the school. ^c^ Education: “Basic schooling” comprises pre-primary, primary and secondary school. “Medium superior level” comprises pre-university school and technical-professional qualifications. “Superior level” comprises bachelor’s degrees, master’s degrees. ^d^ Professionals of special education: the median (range) of work experience with people with ID was 3 (1–12) years.

**Table 2 healthcare-10-01690-t002:** Demographic information of the 69 young females with ID.

	Number (%)
Age, mean (SD)	14 (±9)
**Menstruation**	
Premenarcheal	36 (52)
Menstruating young women	^a^ 32 (46)
Will not have menstruation	1 (1)
**Intellectual disability degree**	
Superficial	9 (13)
Mild	17 (24)
Moderate	15 (22)
Severe	6 (9)
Carers ignored the ID degree	22 (32)
**Aetiology of the ID reported by the carers**	
Multifactorial	50 (72)
Genetic-primary	19 (28)
**Living situation**	
With carer	26 (38)
With family	40 (58)
**^b^ Education level**	
Basic schooling	67 (97)
Medium superior level	1 (1)
Superior level	1 (1)
**^c^ Type of school**	
Public	53 (77)
Private	16 (23)

ID: Intellectual disability. ^a^ Of these 32, their carer reported dysmenorrhoea (*n* = 24), irregular bleeding (*n* = 3), and premenstrual syndrome (*n* = 2). Dysmenorrhoea was managed with pharmacotherapy and homoeopathic medicine. ^b^ Education: “Basic schooling” comprises pre-primary, primary and secondary school, and trade school. “Medium superior level” is a pre-university school. “Superior level” is bachelor’s degree. ^c^ The young women from the Down syndrome clinic attended public school.

## Data Availability

Appendix A: Outline of the topic guide. The dataset used and analysed during the current study is available from the corresponding author upon reasonable request.

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
