# Peer review of "Qualitative Focus Groups with Professionals of Special Education and Parents of Young Females with Intellectual Disability Exploring Experiences with Menstrual Hygiene Management and the Trigger for the Non-Therapeutic Hysterectomy in Mexico"

_healthcare, 2022, doi:10.3390/healthcare10091690_

Round 1
Reviewer 1 Report
I want to tahnk the authors for their tough work in the current manuscript and their contribution to this field of research.
Specific comments.
Comment 1. I propose to revise the title to a shorter version. Please think about it.
Comment 2. Abstract L22-24: Please provide further statistic numbers related to your results.
Comment 3. Introduction Lines 90-91: Please present the main hypothesis of your study if it is available.
Comment 4. Method: Please include a "clear" Figure that illustrtate the study design that you followed.
Comment 5. Please revise the cocnlusion and try to make some proposals related to your findings. What are you proposing to the readers?
Author Response
Please see the attachment.
N.B., at request of the reviewer we have attached a Figure that illustrates the study design. This Figure was not mentioned in the manuscript because we did not consider it necessary. Please see the response to the comment 4 of the reviewer.

Reviewer 2 Report
The objective of the current study was to explore how parents of young females with ID and professionals of special education perceive and experience menstrual hygiene 89 management, which coping strategies are applied; and what triggers hysterectomy performance. There are several issues to clarify.
- Sample and sampling. The sample size was defined by theoretical saturation. What are the indicators of theoretical saturation?
- Accordingly, the analysis involved constant, iterative, and reflexive revisions of each of the narratives, independently completed by at least two researchers (RF and EV), independent of one another; any discrepancy was resolved by consensus. Is triangulation implemented? How to ensure the credibility of data?
- The information generated by means of the focus groups was captured for its analysis and managed by using Atlas ti software (version 8.0. Atlas.ti Scientific Software Development Mnbh, Berlin). The way it is written here makes me very confused as to whether the data was analyzed with analysis software or whether the author summarized the analysis as described earlier in this paragraph.
- The study consisted of a total of 15 focus groups with a mean of 10 ± 5 participants in each group. Fifteen focus group discussions were conducted with a total of 80 participants: 69 parents (61 mothers, four fathers, three grandmothers, and one aunt) of 69 females with ID and 11 professionals of special education. The above two different paragraphs are written in such a way that I am confused, if there are only 80 participants, why 15 groups with an average of 10 participants in each group?
- This study includes parents with moderate and severe intellectual disabilities. How do their views on the issues differ from those of parents with mild intellectual disabilities? I don't seem to see any discussion?
- Menstruation and education. This discussion suggests a more complete discussion of the local situation in Mexico, in addition to a dialogue with the international literature.
- are factors that mirror stigmatized societal views about intellectual disability. Multidisciplinary efforts to reduce the historically conceived inequity in women with ID must be strategically focused on building in-country capacity. This is one of the important core points of this study, and we suggest more discussion and dialogue with the literature.
Reviewer 3 Report
In general, it is a very good job that has all the requirements to be published and be a reference in its field. However, it is necessary to carry out a series of improvements that can bring greater excellence to the whole work. Therefore, my recommendation is that, once it is corrected, it be sent back to the magazine. In general, it is observed that the work is poorly founded, references are scarce both in the introduction and in the methodological part.
The submitted article has an introduction consisting of which the problem is correctly planteated, the justification of its importance and the theoretical foundation prior to the research presented. However, I think it would be very interesting to know in a more developed way the previous research, since it is a very specific topic, and the general knowledge is not so broad. In that sense, I recommend that the authors establish one or two subsections that help to deepen the most relevant theoretical aspects of the research.
In the methodological section, although structured correctly and explained in a meticulous way aspects such as the selection of the sample, the development of the focus groups or their analysis. It is necessary that the part referring to the design of the study be expanded and, in it, explained the reason for the selection of that methodology. If possible, based on previous research of a similar nature. I am surprised that they do not base the thematic analysis on two authors of reference in that analysis, such as Braun and Clark (see references).
In the part of data analysis, we talk about open, selective and axial coding, the three stages of coding of the grounded theory, not of thematic analysis, so this question should be reviewed. The thematic analysis has 5 stages:
1. Familiarization of the data.
2. Generation of catches.
3. Search for core topics or categories
4. Review and verification of topics.
5. Naming and definition of topics
In the results, the part referring to the characteristics of the participants should be in the part where the sample is spoken, it is not a result. The other headings are correctly developed and explained. There is a lack of a semantic network of ATLAS.ti that explains, for example, the connection between the different topics.
The part of the conclusion, if one intends to leave so brief, it would be better to integrate it with the discussion. Personally, I do not think it is necessary since the discussion part is very broad. Likewise, I insist on what I pointed out at the beginning; it must be discussed with more reference than the work currently has.
Braun, V., & Clarke, V. (2006). Using thematic analysis in psychology. Qualitative Research in Psychology, 3(2), 77–101. https://doi.org/10.1191/1478088706qp063oa
Braun, V., & Clarke, V. (2021). Can I use TA? Should I use TA? Should I not use TA? Comparing reflexive thematic analysis and other pattern-based qualitative analytic approaches. Counselling and Psychotherapy Research, 21(1), 37–47. https://doi.org/10.1002/capr.12360
Round 2
Reviewer 2 Report
I are happy with the revised version of your paper.
Author Response
Comments and Suggestions for Authors: I am happy with the revised version of your paper.Response. Thank you for this.
Reviewer 3 Report
Dear Author,
Thank you very much for the changes made. The text of the research acquired important improvements. Before its publication, I would like to make some considerations:
1. The references do not seem to be in the journal format, please check them.
2. Tables 1 and 2 should be in section 2.4 Sample and sampling.
3. Would it be possible to insert Figure 1 in the body of the text?
4. Braun and Clark's references should be cited in the text in addition to being in the references.
Thank you and congratulations for your work
